# How Actin Polymerization and Myosin II Activity Regulate Focal Adhesion Dynamics in Motile Cells

**DOI:** 10.3390/ijms26167701

**Published:** 2025-08-09

**Authors:** Anastasiia Kovaleva, Evgeniya Solomatina, Madina Tlegenova, Aleena Saidova, Ivan A. Vorobjev

**Affiliations:** 1Department of Biology, Lomonosov Moscow State University, Moscow 119991, Russia; a.kovaleva2018@gmail.com (A.K.); solomatinaj@gmail.com (E.S.); 2Engelhardt Institute of Molecular Biology, Russian Academy of Sciences, Moscow 119991, Russia; 3National Laboratory Astana, Nazarbayev University, Kabanbay Batyr Avenue 53, Astana 010000, Kazakhstan; madina.tlegenova@nu.edu.kz (M.T.); aleena.saidova@gmail.com (A.S.); 4School of Science and Humanities, Nazarbayev University, Astana 010000, Kazakhstan

**Keywords:** focal adhesions (FAs), actin polymerization, myosin II, cell adhesion, FA dynamics, cell lamella

## Abstract

Focal adhesions (FAs) are multi-protein complexes that mediate cell attachment to the extracellular matrix. Their formation and maturation depend on intracellular tension generated by actin filaments interacting with phosphorylated myosin II. Using live-cell and confocal microscopy, we investigated how FA dynamics are regulated by actin polymerization and myosin II-driven contractility. We found that knockdown of myosin II resulted in complete and irreversible disassembly of FAs. However, partial inhibition of myosin II, through either ROCK or myosin light chain kinase (MLCK) inhibitors, leads to gradual FA shrinkage. In contrast, complete inhibition of myosin II phosphorylation causes disassembly of existing FAs, followed by the formation of new, small FAs at the cell periphery. In both cases, FAs formed after inhibition of myosin II phosphorylation exhibited significantly longer lifespans than FAs in control cells. Similarly, partial inhibition of actin polymerization using nanomolar concentrations of latrunculin B or cytochalasin D also promoted the formation of small FAs. Complete and irreversible FA disassembly occurred only when actin filaments were fully disrupted, leading to cell lamella retraction. These findings suggest that actin polymerization at the cell edge is the minimal and sufficient requirement for the assembly of small FAs. Notably, our data demonstrate for the first time that perturbation of the actin–myosin system results in stabilization and prolonged lifespan of small FAs, whereas larger FAs, formed in the presence of myosin II activity, are more dynamic. Together, these results emphasize the essential role of cortical actin organization and myosin II phosphorylation in the maintenance and turnover of FAs.

## 1. Introduction

Focal adhesions (FAs) are intricate multi-protein complexes that form at the interface between cells and the extracellular matrix (ECM), serving as key sites for mechanotransduction and cell adhesion [1,2,3,4,5]. These structures are dynamic, undergoing continuous assembly and disassembly, which is tightly regulated by intracellular tension and actin cytoskeletal organization [6,7,8]. The formation of FAs is initiated in the lamella of migrating cells, where integrin clusters interact with the ECM. These early structures, termed nascent adhesions, are initially unstable and either disassemble or mature into larger, elongated FAs through the recruitment of actin-associated proteins and attachment to stress fibers [9,10]. A well-established model suggests that intracellular tension, primarily generated by myosin II-mediated contractility, is essential for FA maturation and stabilization [11,12]. Increased actomyosin-driven tension facilitates the recruitment of mechanosensitive proteins such as talin, vinculin, and paxillin, leading to the enlargement and reinforcement of FAs [12,13]. Conversely, reduced intracellular tension is thought to correlate with FA disassembly and impaired adhesion dynamics [14,15]. However, accumulating evidence indicates that FA assembly can still proceed under conditions of significantly reduced myosin II-mediated tension [16]. For instance, it has been shown that even an 80% reduction in myosin-generated contractility does not completely prevent FA maturation, suggesting the existence of alternative pathways for FA stabilization [17]. Despite extensive research on FA dynamics and the role of myosin II in their maturation, a fundamental question remains unresolved: how do FAs behave when intracellular tension is suppressed, yet the cell’s lamella remains spread on the substrate? Previous studies have primarily examined FA disassembly upon partial inhibition of myosin II or its upstream regulators [12,18,19], but the consequences of complete myosin II inactivation on FA turnover remain controversial [20]. Additionally, while it is well documented that cortical actin polymerization is critical for FA formation and maintenance [21,22], its role in FA remodeling in the absence of myosin II activity remains poorly elucidated. Here, we address these gaps and show, using time-lapse live cell imaging, that partial inhibition of myosin II—either by blocking ROCK kinase or myosin II light chain kinase (MLCK)—leads to gradual shrinkage of focal adhesions, while at the same time increasing their stability Unexpectedly, complete inhibition of myosin II activity induces a complex process of FA’s remodeling: existing FAs disassemble, while new small FAs form predominantly at the cell periphery. Furthermore, we demonstrate that partial suppression of actin polymerization via cytochalasin D or latrunculin B treatment leads to shrinkage of FAs making them at the same time more stable, while rapid and irreversible disassembly of FAs happens only when lamellas are retracted under the action of micromolar concentrations of these actin inhibitors. Our findings suggest that short actin bundles continue to assemble in the lamella despite myosin II inhibition and are sufficient to support the formation of new FAs at the cell edge. However, when actin polymerization is completely blocked, FA formation is entirely abolished. Our study provides new insights into the interplay between actin polymerization and intracellular tension in FA dynamics and highlights the existence of tension-independent mechanisms of FA formation and remodeling.

## 2. Results

### 2.1. Morphology and Dynamics of FAs in Control Cells

3T3 mouse fibroblasts, osteosarcoma U2OS, and A549 lung cancer cells cell lines have been widely used in studies of cell adhesion, migration, and cytoskeletal dynamics [23,24]. All three cell lines form prominent FAs that are easily visualized and could be quantitatively evaluated.

In all three cell types, FAs typically exhibit an ellipsoidal or sometimes elongated shape (Figure 1A–C). In control cells, dynamic FAs were located mainly at the cell edge, and some relatively stable FAs were located in the internal areas. This is in accord with previously described fibrillar adhesions located in the internal cell areas that were less dynamic and contained fewer FA proteins [25,26]. The size and shape of individual FAs were characterized by area and integral brightness (both parameters were taken at the frame with maximal area for a given FA).

FAs in U2OS cells had a median area of 0.76 µm^2^ (0.09–8.48, *N*= 1396) and a median brightness of 2510 a.u. (111–123,682, *N*= 1396). In 3T3 cells, dynamic FAs were distributed more regularly within the cytoplasm and had a median area of 0.86 µm^2^ (0.08–5.45 µm^2^, *N* = 345) and a median brightness of 2823 a.u. (276–61,131 a.u. *N* = 345) (Figure 1D,E).

In migrating cells, most of the FAs were assembled de novo near the spreading edge, a part of them disassembled rapidly, while most increased in area and survived for a relatively long time. In control U2OS cells, the median lifetime of FAs was 32.5 min (range 10–190 min), which was similar to 3T3 cells with an FA median lifetime of 33.5 min (range 6–164 min) (Figure 1F).

### 2.2. Focal Adhesions Decrease in Area and Brightness, but Do Not Completely Disappear After Inhibition of Myosin II Phosphorylation

To determine the role of tensile forces in FA formation and behavior, we used myosin II knockdown and then direct and indirect inhibitors of the myosin II activation pathway, namely, blebbistatin, Y-27632, and ML-7. We first determined that knockdown of myosin II in U2OS cells by specific siRNA resulted in the disassembly of FAs. Transfection with *MYLK*-targeting siRNA (20 nM) resulted in 87.9 ± 12.5% mRNA reduction (*p* < 0.05 vs. scrambled control; Appendix A), confirmed by the loss of protein expression (Appendix A). Microscopic analysis revealed that 82% of transfected cells (41 of 50) exhibited complete dissolution of focal adhesions, while in 9 cells, residual FAs were visualized as small dots (Appendix A).

To further analyze the dynamics of the FA’s response to the decreased myosin II activity, we used specific inhibitors of myosin II phosphorylation and described the effect of these inhibitors on the phosphomyosin distribution and overall level. Using immunofluorescent microscopy for both 3T3 and U2OS cells, we observed the time-dependent decrease of phosphorylated myosin II under these treatments: phosphorylated myosin within 30 min disappeared from the cell edges while it remained in the central part of the cell associated with residual actin fibers. After 3 h of treatment with all three inhibitors, phosphomyosin disappeared almost completely (Appendix A). Western blot analysis of myosin light chain (MLC) phosphorylation levels revealed that the most significant reduction in phosphorylation level occurs at 180 min following treatment with inhibitors of the myosin II activation pathway (Figure 2A,B). At 30 min, the phosphorylation level remains relatively high, showing only a slight decrease, whereas by 180 min, it drops to 3% of the control level for both Y-27632 and blebbistatin. By 360 min, phosphorylation levels do not reach zero; instead, they remain higher than after 180 min treatment, suggesting the involvement of compensatory mechanisms or partial recovery of myosin phosphorylation. For actin polymerization inhibitors latrunculin A and cytochalasin D, no significant reduction in myosin II phosphorylation was observed, indicating that the activation cascade of myosin II does not directly depend on actin dynamics.

Analysis of fixed specimens of cells expressing vinculin-RFP demonstrates that in 3T3 cells, treatment with 45 μM blebbistatin leads to a noticeable reduction in FA size and brightness within 30 min (Figure 3A), with a similar effect persisting after 3 and 6 h (Figure 4E,F). Treatment with 10 µM Y-27632 also causes a rapid decrease in FA size and brightness, although small vinculin-positive adhesions remain at the cell periphery even after 3 and 6 h (Figure 4A,B). Likewise, treatment with 10 µM ML-7 results in a gradual reduction of FA size and brightness over time (Figure 4C,D). In U2OS cells, treatment with 45 μM blebbistatin results in the presence of small, dim focal adhesions after 30 min (Figure 3B), followed by a further decrease in both area and brightness of these FAs after 3 and 6 h treatments (Figure 4K,L). A comparable effect is observed after treatment with 10 µM Y-27632 (Figure 4G,H). However, treatment with 10 µM ML-7 in U2OS cells does not noticeably affect FA area or brightness within the first 30 min, but a decrease in FA area becomes apparent after 3 to 6 h of treatment (Figure 4I,J).

Quantitative analysis of FAs in fixed cells (Figure 4) confirmed that the median FA area decreased significantly over time. Specifically, in 3T3 cells after treatment with 10 µM Y-27362 for 30 min, the area dropped from 0.44 µm^2^ (range 0.09–2.13, *N* = 100) to 0.17 µm^2^ (range 0.04–0.51 µm^2^, *N* = 100); after 3 h, it decreased to 0.12 µm^2^ (range 0.04–0.29 µm^2^, *N* = 100); and by 6 h, it reached 0.10 µm^2^ (range 0.04–0.45 µm^2^, *N* = 100) (Figure 4A). Simultaneously, FA brightness dropped 3.5-fold: from 4849 a.u. (range 203–24,136, *N* = 100) to 1418 a.u. (range 306–9419, *N* = 100) in 30 min, followed by a slower decrease to 909.5 a.u. (range 130–5150, *N* = 100) and 1047 a.u. (range 184–6050, *N* = 100) at 3 and 6 h, respectively (Figure 4B). In U2OS cells treated with Y-27632, we also observed a progressive decrease in both FA area and brightness, although more gradually (Figure 4G,H). Differences between control and Y-27632-treated cells were statistically significant at all time points (Kruskal–Wallis test with Dunn’s post test, *p* < 0.001).

A similar trend was observed in 3T3 cells treated with ML-7. FA area gradually decreased from 0.44 µm^2^ (range 0.09–2.13, *N* = 100) to 0.29 µm^2^ (range 0.08–1.34 µm^2^, *N* = 100) within 30 min, and further decreased to 0.18 µm^2^ (range 0.04–0.71 µm^2^, *N* = 100) and 0.19 µm^2^ (range 0.05–0.85 µm^2^, *N* = 100) after 3 and 6 h, respectively (Figure 4C). Correspondingly, FA brightness decreased from 4849 a.u. (range 203–24,136, *N* = 100) to 2652 a.u. (range 303–24,189, *N* = 100) within 30 min, followed by a gradual reduction to 1540 a.u. (range 256–11,930, *N* = 100) by 3 h and remained at a similar level after 6 h (Figure 4D). U2OS cells treated with ML-7 showed a similar pattern of gradual FA fading and area reduction (Figure 4I,J). However, for ML-7 treatment of U2OS cells, large differences in FA area and brightness compared to the control were observed only at the 3- and 6-h time points (Kruskal–Wallis test with Dunn’s post test, *p* < 0.01).

Blebbistatin treatment caused a rapid decline in both FA area and brightness in both cell lines. In 3T3 cells, FA area decreased sharply within 30 min from 0.44 µm^2^ (range 0.09–2.13, *N* = 100) to 0.16 µm^2^ (range 0.04–0.6, *N* = 100), followed by a slower reduction and stabilization at 0.13 µm^2^ (range 0.04–0.45, *N* = 100) after 3 h (Figure 4E). Similarly, FA brightness dropped from 4849 a.u. (range 203–24,136, *N* = 100) to 2279 a.u. (range 253–11,619, *N* = 100) in 30 min, then reached a plateau with values of 1392 a.u. (range 200–10,404, *N* = 100) after 3 h, remaining nearly unchanged after 6 h (Figure 4F). U2OS cells showed the same pattern of rapid FA reduction within the first 30 min, followed by stabilization at 3 and 6 h (Figure 4K,L). For all time points, differences between the control and blebbistatin-treated cells were statistically significant (Kruskal–Wallis test with Dunn’s post test, *p* < 0.001). 

Thus, we observed the two phases in the behavior of FAs under treatment with myosin II inhibitors. The first one lasts from 30 min to 1 h after the addition of the inhibitor when the cells are in a transition state, and we observe a disassembly of FAs and a rapid decrease in area and brightness for the remaining FA population. The second phase begins between one and three h after adding the inhibitor, when cells come to a new steady state, and the population of dim and small FAs could still be detected at the cell edge, both for 3T3 and U2OS cells. Since FAs were still visible after 3 h of treatment, we suggested that either residual FAs were stabilized or new FAs continued to form under the action of the myosin II inhibitor. To test this, we performed a detailed analysis of FA dynamics at different time points after myosin II inhibition.

To confirm that FA measurements in cultured cells under the action of myosin II inhibitors are not affected by fixation procedure, we imaged RFP-vinculin in blebbista-tin-treated live cells and analyzed dynamics of FAs in the cell lamellum. All measure-ments before and after addition of inhibitor were performed in uniform conditions and for the same cells. For all inhibitors decrease in FA area and brightness values were similar when measured on the fixed specimens and on time-lapse frames (Appendix A).

The most prominent changes in FAs were observed under the treatment of blebbistatin. New lamellipodia were formed in 3T3 cells within 30 min after adding the inhibitor. Also, existing lamellae did not shrink significantly even after 3 h. The total number of FAs per cell decreased by 4 to 5 times. Later on (after one h), small and pale FAs were present at the cell edge, and few FAs could be observed in the cell interior. In U2OS cells, the total number of FAs per cell also decreased 4 to 5 times, but unlike for 3T3 cells, FAs in the cell body completely disassembled and remained only near the cell edges.

To assess the dynamics of FAs at the cell edge in the steady state where phosphorylated non-muscle myosin II (NMII) was completely absent, we performed time-lapse imaging and analyzed the lifetimes of FAs that formed after 195 min from the initial frame have been taken (i.e., 3 h after inhibitor addition). The absence of phosphorylated NMII at this time point was confirmed by Western blot and immunohistochemistry (Figure 2 and Appendix A), ensuring that only de novo FAs formed in the absence of active NMII were included in the analysis.

For all three myosin II inhibitors tested, we observed a statistically significant increase in the lifespan of these newly formed FAs compared to control cells (Figure 5A,B). In control 3T3 cells, the median FA lifetime was 33.5 min (range 6–164 min). Upon blebbistatin treatment, the median FA lifetime approximately doubled to 62.5 min (range 20–280 min, Kruskal–Wallis test, *p* < 0.05). Treatment with ML-7 resulted in a median FA lifetime of 59.5 min (range 18–177 min), while Y-27632 increased the median to 49.5 min (range 13–121 min). For all three treatments, the difference from control was statistically significant (Kruskal–Wallis test, *p* < 0.05).

In U2OS cells, the response pattern differed slightly. The greatest increase in FA median lifetime was observed following ML-7 treatment (70 min, range 20–245 min), followed by Y-27632 (57.5 min, range 15–190 min), and blebbistatin (40 min, range 15–245 min), compared to 32.5 min (range 10–190 min) in control. As in 3T3 cells, the differences in FA lifetime between control and treatment groups were statistically significant for all three inhibitors (Kruskal–Wallis test, *p* < 0.01) (Figure 5B).

### 2.3. Blebbistatin Treatment Promotes Disassembly of Pre-Formed FAs and Inhibits Growth of New FAs

At the same time, we observed the formation of new FAs in both cell lines under blebbistatin treatment. To analyze whether blebbistatin affects the dynamics of these new FAs, we compared the dynamics of the area and integrated brightness of FAs before and after the addition of the inhibitor in the same cells at higher temporal resolution (1 frame per min) and detected the significant decrease in the assembly and disassembly times of those adhesions. In control cells, the phase of FA assembly continued for 6 min (median, *N* = 25). Under the blebbistatin treatment, the FA assembly phase shortened to 2 min (median, *N* = 20). While in the control, the FA area increased during the growth period 3.5-fold, and in the presence of blebbistatin, the area increase was only 2-fold (Figure 5C), with the same difference in brightness dynamics (Figure 5D). Under blebbistatin treatment, the median fluorescence intensity of mature FAs was four-fold lower than in control cells—2566 A.U. instead of 10,044 A.U. (*p* < 0.00003) (Figure 5E). Thus, inhibition of myosin light chain leads to a shortening of the FA growth period, a decrease in the total amount of fluorescent vinculin within an individual FA, and results in a significant decrease of vinculin density in the FA.

Under blebbistatin treatment, the disassembly of pre-existing FAs occurs either simultaneously with or slightly preceding the formation of new ones. The emergence of newly formed FAs during blebbistatin treatment was confirmed by time-lapse microscopy initiated immediately after blebbistatin was added to A549 cells (Appendix A). Within 20–30 min, all pre-existing FAs were disassembled. Over the following 30 min, a transient population of small, short-lived FAs appeared in the central part of a cell and at the cell periphery. Small FAs continued to form at the cell edge for at least 2 h after treatment. This indicates that, during the first hour of blebbistatin exposure, the entire FA population is in a transitional phase.

To examine this transition phase in greater detail, we analyzed FA dynamics by measuring their area and integrated intensity in the same U2OS cells (*N* = 5) before and after blebbistatin treatment. Quantitative measurements were taken in three frames prior to drug addition, then every 5 min during the first 30 min post-treatment, and at additional time points: 45 min, 2.5 h, 3 h, and 3.5 h. In U2OS cells, both the FA area and integrated intensity remained stable before inhiwebitor addition (Kruskal–Wallis test with Dunn’s test, *p* = 0.36). However, both parameters showed a significant decline within the first 5 min after blebbistatin addition and continued to decrease until reaching a plateau by 45 min (Kruskal–Wallis test with Dunn’s test, *p* < 0.0001) (Figure 5F,G).

These observations indicate that the reduction of contractile tension during the transition phase promotes disassembly of existing FAs but does not prevent the formation of new, small FAs.

### 2.4. Actin Cytoskeleton Changes Under the Action of Myosin II Inhibitors

In all cell lines examined, after 3 h of blebbistatin treatment, cells remained well spread on the substrate; however, actin cytoskeleton organization was markedly altered (Figure 6). Actin predominantly accumulated at the extreme periphery of the cell, while phosphorylated myosin displayed diffuse staining in the central cytoplasm and was no longer associated with actin cables (Appendix A). Stress fibers and other well-organized F-actin structures, such as dorsal arcs and peripheral arcs, were greatly diminished. In the membrane-proximal regions enriched with actin, small focal adhesions were observed, although they were not associated with prominent actin bundles. In some U2OS cells, stress fibers were almost completely disassembled (Figure 6A). In contrast, 3T3 cells retained some shortened stress fibers and scattered small FAs in the central region of the cell body (Figure 6B). Given the dramatic reorganization of the actin cytoskeleton following inhibition of myosin II phosphorylation, we next examined the behavior of focal adhesions in the presence of specific inhibitors of actin polymerization.

### 2.5. Actin Filaments Are Necessary for the Formation of New Dynamic FAs

To test whether FA formation on the cell edge depends on the presence of actin bundles, we first treated cells with 1µM of cytochalasin D or 1 µM of latrunculin B and visualized actin, phosphomyosin II (Figure 7), and vinculin (Appendix A) distribution using confocal microscopy. At least 40 cells were examined at each time point.

The 30-min incubation of 3T3 fibroblasts and U2OS cells with cytochalasin D caused the disassembly of all actin and phospho-myosin II structures in 80% of cells; in the remaining 20%, thin actin fibers remained on stable edges associated with a small amount of phosphorylated myosin. After 3 and 6 h under cytochalasin D treatment, actin fibers were disassembled in all cells observed, and only punctate staining of phospho-myosin II remained. Similarly, latrunculin B caused the disassembly of the F-actin cytoskeleton, and only punctate phospho-myosin remained in all cells within 30 min. After 6 h of incubation with the inhibitors, we observed partial shrinkage of cells (Figure 7A,B). However, small FAs remained even after 6 h treatments (Appendix A).

To test in more detail whether FA formation requires residual actin structures, A549 cells were treated for 2 h with increasing concentrations of cytochalasin D (100 nM–10 μM) or latrunculin B (300 nM–3 µM).

Treatment with 100 nM of cytochalasin D results in partial shrinkage of cells and disassembly of long stress fibers in the majority of cells, while short actin fibers remained at the cell edges and numerous FAs were present at the cell margins (Figure 8B). At the increasing concentrations of cytochalasin D, the number of FAs decreased, yet some could be observed at the cell margin after treatment with 1 μM of the drug (Figure 8C). Higher concentration of cytochalasin D (3–10 μM) resulted in the complete shrinkage of the cell lamella, and only residual cytoplasmic staining of paxillin was observed in some cells (Figure 8D).

Treatment with latrunculin B results in similar outcomes. After incubation with 300 nM latrunculin B, lamellas in all cells partially shrink, and in 85% of cells, stress fibers disassembled completely, while in other cells, residual stress fibers remained (Figure 8E). Small FAs were observed at the cell edges in half of the cells.

Treatment with higher concentrations of latrunculin B results in complete shrinkage of lamellas and rounding of cell bodies. After 1 μM latrunculin, residual FAs were present in 28% of cells, while only diffuse staining in the cytoplasm was observed in other cells (Figure 8F). After incubation of cells with 3 μM latrunculin B, the effect on cell morphology was the same; however, there were no FAs at all (Figure 8G).

Based on the results obtained on three cell cultures, we conclude that partial disassembly of F-actin when lamellae remain results in preservation of small FAs at the cell periphery. Only high concentrations of actin polymerization inhibitors, leading to complete retraction of the lamella, are sufficient for complete disassembly of FAs.

## 3. Discussion

Development of FAs in motile cells is a multistep process. First, short-living nascent adhesions are rapidly formed at the advancing cell edge. Then, tension of actin fibers near the cell edge generated by phosphorylated myosin II located deeper in the cytoplasm promotes growth and maturation of FAs with a lifespan of 10 min and more [27]. Under the tension, FAs enlarge and exhibit a distinct composition, characterized by the accumulation of mechanosensitive proteins such as talin, vinculin, and paxillin [28,29,30]. Reduced intracellular tension is thought to lead to the breakdown of FAs and prevent the formation of new ones due to increased dissociation of mechanosensitive proteins [15,31]. Cortical actin supports only the early stages of FA assembly [2,16,32,33]. Together with myosin II, these actin structures regulate the formation, stability, and function of FAs and thus facilitate essential processes such as cell adhesion, spreading, polarization, migration, and mechanosensing [33,34,35]. It is generally accepted that intracellular tension is generated by the interaction of actin fibers with phosphorylated myosin II and is applied to the inner layer of FAs through actin fibers originating there [36]. The question is, what is the correlation between tensile force and the size of FAs? There are some cues that strong myosin II-mediated tension is not critical for FA formation, at least at the initial stages of maturation [37,38]. The compositional maturation of FAs through recruitment of adhesion proteins and matrix remodeling occurs even when the myosin-mediated tension cues are reduced by 80%, though a minimal level of myosin II activity is required to facilitate the formation and maturation of FAs at the leading edge [16,39].

In the current study, we analyzed the relationship between the formation, growth, and disassembly of FAs in relation to myosin II activation and actin polymerization by using specific inhibitors. All three inhibitors (Y27632, ML-7, and blebbistatin) of the myosin II activation pathway led to significant reductions in the FAs. Detailed live cell visualizations of blebbistatin-treated cells show that the response to the decreased tension is complex: existing FAs shrink within 10–20 min, but later on, a new steady state is achieved, and even new FAs can form in the presence of the inhibitor. These observations confirm that apparently stable FAs are indeed highly dynamic, i.e., their size depends on the assembly–disassembly equilibrium rather than the scaffold. Rapid disassembly of FAs after addition of myosin II inhibitors is in accord with the results of FRAP experiments, demonstrating relatively short half-life times for the major FA proteins, including FAK, paxillin, and vinculin [15,40,41,42]. However, after prolonged treatment with myosin II inhibitors, incompletely disassembled FAs enter into the new steady state when their turnover is slowed down, and even new small FAs are formed. We suggest that in the absence of myosin II phosphorylation, the first stage of FA assembly, including vinculin recruitment, still happens, but in the absence of tension, this process is limited and subsequent growth of FAs cannot occur.

Alterations in intracellular tension and focal adhesion (FA) dynamics observed upon myosin II inhibition are likely to reverberate through key signaling cascades mediated by focal adhesion kinase (FAK) and Src. Under normal contractile conditions, FAK undergoes autophosphorylation at Tyr397, creating a high-affinity binding site for Src’s SH2 domain; this FAK–Src complex then phosphorylates scaffold proteins such as paxillin on Y31/118, recruiting additional effectors that activate MAPK and PI3K–Akt pathways to regulate cell migration and survival [18]. Loss of FAK–Src complex formation would be expected to diminish paxillin phosphorylation, attenuate Rac1 and ERK1/2 activation, and impair feedback on RhoA-mediated cytoskeletal remodeling, thereby reinforcing the stabilization of small, non-maturing adhesions that we observe.

Overall, it is suggested that the necessary and sufficient structures responsible for the formation of dynamic nascent adhesions in the cells under complete loss of myosin II-dependent tension are polymers of cortical actin that continue assembling under the plasma membrane [42,43]. Confirming this supposition, we observed the presence of a continuous layer of F-actin fibers near the cell edge after blebbistatin treatment (Figure 7). To test the role of the cortical F-actin layer directly, we treated cells with the inhibitors of actin polymerization (latrunculin B and cytochalasin D) and observed a dose-dependent effect on disassembly of actin fibers under both cytochalasin D and latrunculin B treatments. Using phalloidin staining and high-resolution confocal microscopy, we confirmed that the presence of a continuous layer of F-actin fibers near the cell edge remained in the presence of nanomolar concentrations of actin inhibitors, while this layer was completely lost after treatments with high doses of latrunculin B or cytochalasin D.

In accord with the previous data [21,43], we suggest that short actin bundles are still growing under the action of blebbistatin or moderate doses of cytochalasin D or latrunculin B, thus allowing the formation of small FAs. The possible mechanisms underlying the fine interaction between cortical actin and FAs may include the Arp2/3 complex interacting with vinculin and FAK [44,45]; VASP binding to actin and associating with vinculin and zyxin [46]; direct binding of vinculin and talin to actin [47,48]; and α-actinin, an actin cross-linker in lamellipodia [49]. Alternative mechanisms may involve mDia2, a formin involved in actin polymerization, which was shown to be localized to the cortical actin network in migrating cells. This localization of actin-binding proteins is essential for maintaining polymerization-competent actin filaments at FAs, thus playing a significant role in FAs’ dynamics [43,50,51].

While a large body of evidence supports the idea that cortical actin supports the early stages of FA assembly, it is usually suggested that small FAs are always ephemeral and cannot be stable. Our observation clearly shows that small FAs growing in less than 2 min under continuous blebbistatin treatment are even more stable than larger ones in untreated cells, with a lifespan of about 1 h.

Overall, we demonstrate that partial inhibition of myosin II activity or disruption of actin stress fibers results in a decrease in FA size but stabilizes them. Complete inhibition of myosin II results in the rearrangement of the FA array and formation of new small FAs, which do not enlarge; however, they are relatively stable, keeping lamella attached to the substrate. Inhibition of actin polymerization results in complete and irreversible disassembly of FAs only in the case when lamella retracts completely (Figure 9). These insights advance our understanding of the mechanisms underlying cell spreading, polarization, and migration, particularly the role of the cortical actin network in these processes.

## 4. Materials and Methods

### 4.1. Cell Culture

Model cell lines 3T3 (mouse fibroblasts) and U2OS (human osteosarcoma cells) (ATCC, Manassas, VA, USA) transduced with vinculin-RFP gene were cultured in a 1:1 mixture of DMEM and F12 media (Sigma Aldrich, St. Louis, MO, USA) supplemented with 5% fetal calf serum (PanEco, Moscow, Russia), 2 mM glutamine (PanEco, Moscow, Russia), and 90 μg/mL gentamicin (PanEco, Moscow, Russia) at 37 °C in a humidified atmosphere with 5% CO_2_. A549 cells (ATCC, Manassas, VA, USA) were grown in Dulbecco’s modified Eagle medium (DMEM) (Thermo Fisher Scientific, Waltham, MA, USA) supplemented with 10% fetal bovine serum (FBS) (Thermo Fisher Scientific, Waltham, MA, USA), 4–8 mM of L-glutamine (Sigma Aldrich, St. Louis, MO, USA), and penicillin–streptomycin mixture (Sigma Aldrich, St. Louis, MO, USA) at 37 °C in a humidified atmosphere with 5% CO_2_. Transient transfection of A-549 cells with Ptag-RFP-vinculin vector (Eurogen, Moscow, Russia) was carried out using X-treme GENE HP DNA transfection reagent (Roche, Basel, Switzerland) according to the manufacturer’s instructions.

### 4.2. Production of 3T3 and U2OS Model Cell Lines with Stable Vinculin Expression

To generate 3T3 and U2OS cell lines with stable vinculin-RFP expression, vinculin cDNA was cloned into the pSLIK lentiviral vector containing the RFP gene as a reporter and a puromycin resistance gene for selection (Evrogen, Moscow, Russia). HEK293T cells (ATCC, Manassas, VA, USA) were transfected with the lentiviral vector using X-TremeGENE HP DNA transfection reagent (Sigma Aldrich, St. Louis, MO, USA) following the manufacturer’s protocol. Viral particles were harvested from the medium at 24 and 48 h after transfection. To evaluate transduction efficiency, HEK293T cells were infected with serial dilutions of the viral particles. To enhance infection efficiency, the cationic polymer polybrene (Sigma Aldrich, St. Louis, MO, USA) was added to the culture medium at a final concentration of 8 μg/mL. After 48 h of transduction, the number of transducing units (TU) per ml was calculated by flow cytometry based on the percentage of RFP-expressing cells.

For the infection of model cell lines, 5 × 10^6^ 3T3 and U2OS cells were seeded in 2 mL of culture medium and infected with viral particles. The minimum TU concentration was at least 10^7^ TU/mL. After 24 h, the medium was changed. Infected cells were cultured in the full culture medium containing 0.1% puromycin (Gibco, Carlsbad, CA, USA). After 5 days, cells were sorted by RFP fluorescence intensity (excitation 561 nm, emission 585/15 nm BP) using a FACSAria SORP cell sorter (BD Biosciences, San Jose, CA, USA) with an 85 μm nozzle and corresponding pressure parameters. After obtaining cell lines with stable vinculin-RFP expression, 3T3-RFP or U2OS-RFP cells with medium fluorescence levels of the RFP (27% of events with MFI 168) were sorted, excluding cells with high RFP fluorescence.

### 4.3. Live Cell Fluorescence Microscopy

Model cell lines were detached using a trypsin-EDTA solution (PanEco, Russia) and seeded in 4-well glass-bottom Petri dishes (35 mm) (Cellvis, Mountain View, CA, USA). After 24 h, an experimental wound was created in the monolayer using a 10 μL pipette tip. Cell debris was removed by washing twice with serum-free DMEM. To maintain cell viability, the medium was replaced with a 1:1 mixture of DMEM/F12 and CO_2_-independent culture medium (Gibco, USA), overlaid with mineral oil (Sigma, USA). Time-lapse imaging was performed using an inverted Nikon Eclipse TiE microscope (Nikon, Tokyo, Japan) with a phase-contrast Plan Apo 60×/1.4 objective (Nikon, Japan). Images were recorded using an ORCA-Flash 4.2 CMOS camera (Hamamatsu, Shizuoka-ken, Japan) controlled by MicroManager software (version 2.0.0 (13 July 2021) (https://micro-manager.org/)). The exposure time was 500 ms for fluorescence and 100 ms for phase-contrast images. The equivalent pixel size was 0.108 µm. Imaging was performed at 5-min intervals for 6.5 h at 37 °C. For analysis of FAs in the same cells before and after cytoskeleton inhibitor treatment, inhibitors were added 30 min after the start of time-lapse imaging, and recording was performed with 2-min time intervals for 3 h.

### 4.4. Immunocytochemical Staining

Cells were fixed with 4% paraformaldehyde in PBS (pH = 7.2–7.4) for 20 min at room temperature and washed three times with buffer and then permeabilized with 0.1% Triton-X-100 (Amresco, Solon, OH, USA) and 0.01% Tween-20 (Amresco, Solon, OH, USA) for 20 min. After three washes with PBS, cells were incubated with primary rabbit antibodies against bisphosphorylated myosin II (Thr18/Ser19) (Invitrogen, Carlsbad, CA, USA) at a 1:200 dilution overnight at 4 °C and with secondary anti-rabbit antibodies conjugated with Alexa-488 (ab150077, Abcam, Cambridge, UK) at 37 °C for 60 min. After washing, actin microfilaments were stained for 30 min with Phalloidin-Alexa-647 (ab #176759, Abcam, Cambridge, UK). Images were acquired using an inverted Zeiss Axio Observer microscope with a Colibri 7 LED light source and multiple filter sets for GFP, m-Kate, and AlexaFluor647 (Carl Zeiss, Oberkochen, Germany, Cat. No 489090–9110) with a 63×/1.46 Plan Apo oil immersion objective. Images were recorded using an ORCA-Flash 4 V2 sCMOS camera (Hamamatsu, Shizuoka-ken, Japan) controlled by Zeiss Zen Blue 3.1 software. For each condition, at least 10 fields of view were imaged.

### 4.5. siRNA Transfection and Validation

siRNA targeting the human myosin light chain kinase (*MYLK*) gene (NCBI RefSeq: NM_053025.4) was designed using the BLOCK-iT™ RNAi Designer tool (Thermo Fisher Scientific; https://rnaidesigner.thermofisher.com/rnaiexpress/ (accessed on 4 June 2025)). The selected target sequences (CATCACCCTGCAGGCCGCAAGAGTT, CAGAGGGCAAGAAGCTGCTGCTCCA, and GGTCCAGTGTGTGGATGCCTTTGAA) were chosen based on algorithmic recommendations and further validated for specificity using -NCBI BLAST+ version 2.13.0 (https://blast.ncbi.nlm.nih.gov/Blast.cgi, accessed on 4 June 2025) against the human transcriptome. siRNA duplexes, including the non-targeting scrambled siRNA, were synthesized and purified by HPLC (OBio Technology, Shanghai, China) and reconstituted in nuclease-free water at a stock concentration of 20 µM. All siRNA duplexes were verified for integrity by mass spectrometry and denaturing PAGE, aliquoted, and stored at −80 °C.

For siRNA delivery, U2OS cells were transfected with 20 nM *MYLK*-targeting siRNA using X-tremeGENE™ siRNA Transfection Reagent (Sigma Aldrich, St. Louis, MO, USA) according to the manufacturer’s protocol. Briefly, siRNA was diluted in serum-free Opti-MEM^®^ I (Gibco, Carlsbad, CA, USA) and combined with X-tremeGENE™ reagent at a 1:1 ratio (final volume per well 100 µL). The mixture was incubated for 20 min at room temperature, then added dropwise to the cells. Cells were incubated with the transfection complexes under standard culture conditions for 48 h before subsequent analyses.

### 4.6. RTqPCR

Total RNA was isolated from transfected U2OS cells using the RNeasy Mini Kit (Qiagen, Venlo, The Netherlands) according to the manufacturer’s protocol, including on-column DNase I digestion to eliminate genomic DNA contamination. RNA concentration and purity were determined spectrophotometrically (NanoDrop, Thermo Fischer Scientific, Waltham, MA, USA; A260/A280 > 1.9). First-strand cDNA synthesis was performed with 500 ng total RNA using the iScript™ cDNA Synthesis Kit (Bio-Rad, Hercules, CA, USA) under the following conditions: 25 °C for 5 min (primer annealing), 46 °C for 20 min (reverse transcription), and 70 °C for 1 min (enzyme inactivation).

Quantitative real-time PCR was carried out using SsoAdvanced™ Universal SYBR^®^ Green Supermix (Bio-Rad, Hercules, CA, USA) on a CFX96 Touch™ system (Bio-Rad, USA). *MYLK*-specific primers (forward 5′-TTCAGGACGCCTTTCCTGTC-3′, reverse 5′-GGGCAAAATGAAAGCAGGGG-3′, product length 179 bp, Ta 62 °C), *GAPDH*-specific primers (forward 5′-TGCACCACCAACTGCTTAGC-3′, reverse 5′-GGCATGGACTGTGGTCATGAG-3′, product length 87 bp, Ta 60 °С) were used. Reaction mixtures (20 µL final volume) contained 10 µL SYBR Green Master Mix, 0.5 µM of each primer, and 2 µL of diluted cDNA. Thermocycling parameters: initial denaturation at 95 °C for 3 min; 40 cycles of 95 °C for 15 s and Ta°C for 30 s. Melt curve analysis (65–95 °C, increment 0.5 °C/5 s) confirmed single-product amplification. *MYLK* expression was normalized to the *GAPDH* gene expression using the 2^(−ΔCq)^ method. Statistical significance between the siRNA-treated group and control group was assessed by unpaired Student’s t-test (GraphPad Prism v9.0; *p* < 0.05 considered significant).

### 4.7. Inhibitory Analysis

Myosin II ATPase activity was inhibited using 45 μM blebbistatin (Adooq Bioscience, Irvine, CA, USA). ROCK and MLCK kinases were inhibited using 10 μM Y-27632 (Sigma, USA) and 10 μM ML-7 (Sigma, USA), respectively. Actin polymerization was inhibited using cytochalasin D (Sigma, USA) or latrunculin B (Calbiochem, San Diego, CA, USA).

### 4.8. Confocal Microscopy with Airyscan Super-Resolution Module

To visualize the residual actin filaments in U2OS cells after 3 h of myosin II ATPase activity inhibition (45 μM blebbistatin), immunocytochemical staining was performed as described above. Cells were stained with rabbit antibodies against paxillin (Ab #32084, Abcam, UK) and secondary anti-rabbit antibodies conjugated to AlexaFluor 488 (Ab #150077, Abcam, UK). Phosphorylated myosin was stained with rabbit antibodies against myosin II (Thr18/Ser19) (Invitrogen, USA) and secondary anti-rabbit antibodies conjugated to AlexaFluor 568 (Ab#175692, Abcam, UK). All antibodies were used at a final dilution of 1:200. Actin microfilaments were stained with Phalloidin-Alexa-647 (Ab#176759, Abcam, UK) at a final dilution of 1:50. Chromatin was stained with DAPI at a concentration of 2 μg/mL. Images were acquired using a Zeiss LSM780 confocal microscope or a Zeiss LSM900 confocal microscope with an Airyscan super-resolution module. Scanning was performed with a 0.2 μm step along the Z-axis in “Frame” mode, with a pixel size of 0.035 × 0.035 μm in the X-Y plane. Detector gain was set automatically. For fluorescence imaging, lasers with wavelengths of 488 nm, 561 nm, 647 nm, and 405 nm were used. Final images were obtained by summing optical sections by maximal intensity projection (MIP) in ImageJ software version 1.53t (https://imagej.net/ij/, accessed on 4 June 2025).

### 4.9. Western Blot

Cell pellets of 3T3 and U2OS model lines were lysed in RIPA buffer containing 1× protease inhibitor cocktail (Sigma Aldrich, St. Louis, MO, USA) and phosphatase inhibitors (Roche, Switzerland). Lysates were incubated for 30 min at 4 °C and clarified by centrifugation at 12,000 rpm for 20 min. Supernatants were transferred to clean tubes, and total protein concentration was measured using a spectrophotometer (Nanodrop, Thermo Fischer Scientific, USA). The volume of protein extract required to load 40 μg per well was calculated and adjusted accordingly. Samples were mixed with 5× Laemmli sample buffer and denatured by heating at 100 °C for 5 min. Equal amounts of protein were loaded onto a 12.5% SDS–polyacrylamide gel and resolved by electrophoresis. Proteins were subsequently transferred onto Immobilon-P PVDF membranes (Merck Millipore, Darmstadt, Germany) using standard wet transfer protocols. The membrane was washed three times with 1× TBS-T buffer and blocked with 5% non-fat dry milk (Cell Signaling Technology, Danvers, MA, USA) for 30 min at room temperature. The membrane was incubated overnight at 4 °C with primary antibodies diluted in TBS-T buffer containing 1% BSA. Dilutions used: 1:500 for rabbit anti-phospho-myosin light chain 2 antibodies (Thr18/Ser19, #3674, Cell Signaling Technology, USA), 1:1000 for mouse myosin light chain monoclonal antibodies (AB03/2F2, Bio-Rad Laboratories, USA), 1:200 for mouse anti-α-tubulin antibodies (ab#7291, Abcam, UK).

After primary antibody incubation, the membrane was washed three times with TBS-T buffer at room temperature and then incubated with secondary anti-mouse HRP-conjugated antibodies (#7076, Cell Signaling Technology, USA) and anti-rabbit HRP-conjugated antibodies (#7074, Cell Signaling Technology, USA) diluted 1:1000 in TBS-T buffer containing 5% non-fat dry milk, for 60 min at room temperature. The membrane was washed three times with TBS-T buffer at room temperature. For detection of HRP-conjugated antibodies, the Clarity Western ECL system (BioRad, USA) was used according to the manufacturer’s protocol. Chemiluminescent signals were visualized using the iBright imaging system (Invitrogen, USA).

Western blot quantification was performed using ImageJ software (version 1.54m, https://imagej.net/ij/, accessed on 4 June 2025). For each protein band, a rectangular region of interest (ROI) was selected, and densitometric analysis was conducted after background subtraction using the “Rolling Ball” algorithm to correct for uneven illumination. Band intensity was normalized to the loading control (alpha-tubulin) to account for variations in sample loading, and the ratio of the target protein signal to the loading control signal was calculated for each lane. The relative intensity values were averaged across at least three independent biological replicates, normalized to the control condition, and expressed as fold change. Statistical significance was determined using one-way ANOVA, as appropriate. All results are presented as mean ± standard deviation (SD).

### 4.10. Image Processing and Data Analysis

The 16-bit images were processed using ImageJ software (version 1.54m, https://imagej.net/ij/, accessed on 4 June 2025). Noise reduction and contrast enhancement were performed using brightness-contrast adjustment, background subtraction tools, and difference of Gaussians using the kernel one to kernel two ratio 1:5 (where indicated). FAs were manually outlined using the freehand selection tool, and the resulting contours were saved in the ROI manager. Contours were transferred to unprocessed (RAW) images, and the integral brightness and area were calculated for each FA in each frame. FA area and integral density were determined at the first frame, where both parameters were visually different from the surrounding cytoplasm, and followed until the last frame, where both parameters were visually different from the surrounding cytoplasm. Growth and disassembly periods were determined manually from the life histories of FAs as periods when FA area increased or decreased, consequently, as described elsewhere using linear regression with R^2^ ≥ 0.8 as a threshold (see [52] for details). Data were averaged for at least 20 FAs in control cells and cells treated with each inhibitor.

### 4.11. Statistical Analysis and Data Visualization

Statistical analysis was performed using Prism 9.0 software (GraphPad Software Inc., Boston, MA, USA). For comparison of multiple non-parametric groups, the Kruskal–Wallis one-way analysis of variance with Dunn’s test was used. Statistical significance was set at *p* < 0.05. Data are presented as medians with ranges if otherwise not stated.

## 5. Conclusions

Relatively rapid disassembly of large FAs happens after inhibition of myosin II or after partial inhibition of actin polymerization. In both cases, lamella does not shrink significantly, and small yet rather stable FAs are formed de novo at the cell edges. When actin polymerization is significantly perturbed, FAs disassemble in several minutes, lamella shrinks, and no new FAs form. We propose a model where, after myosin II relaxation or partial actin disorganization, lamella could be supported by small FAs that are able to form but do not enlarge because of the lack of tension due to the disruption of the cytoskeleton. These small FAs turn over more slowly than normal ones and require only the presence of polymerized actin filaments at the cell cortex.

## 6. Limitations of the Study

We acknowledge that the inhibitors used have broader effects beyond altering FAs, particularly on cortical actin organization and overall cytoskeletal remodeling. Myosin II is known to regulate actin network architecture not only by generating contractile forces but also by promoting actin filament disassembly [53,54,55]. Given the distinct mechanisms of myosin light chain inhibition, the non-identical effects of Y-27632, ML-7, and blebbistatin on FAs and the actin cytoskeleton cannot be excluded.

Therefore, while our data strongly suggest that the observed FA changes result from decreased actomyosin contractility, we recognize that concurrent alterations in cortical actin organization may also contribute to these effects. A detailed investigation of how linear and branched actin architectures could be affected by myosin II inhibition in the context of FA dynamics remains an important avenue for future studies.

We also acknowledge the lack of direct measurements of myosin II-dependent mechanical forces within the cells. Future experiments employing direct measurements of intracellular tension, such as traction force microscopy or Förster resonance energy transfer (FRET)-based tension sensors based on talin or vinculin, would provide additional valuable insights and more conclusive assessments on the formation of small stable FAs.

## Figures and Tables

**Figure 1 ijms-26-07701-f001:**
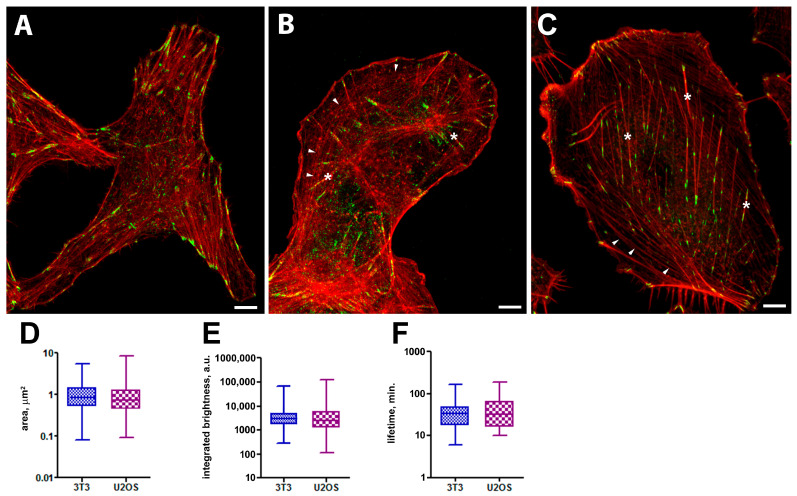
Focal adhesions (vinculin-RFP expression) and F-actin (phalloidin-Alexa 647 staining) in control cells. Airyscan images of vinculin (green) and actin (red) in 3T3 (**A**) and U2OS cells (**B**). (**A**) In 3T3 cells, the actin cytoskeleton mainly consists of stress fibers running along the cell axis or perpendicular to the cell edge. FAs are mainly localized at the cell edges and are often associated with the ends of stress fibers. (**B**,**C**) In U2OS and A549 cells, the actin cytoskeleton also contains dorsal fibers (asterisks) and actin arcs (arrowheads). Area (**D**), integrated brightness (**E**), lifetime (**F**). Data on the graphs are shown as median with range. Scale bar—5 µm.

**Figure 2 ijms-26-07701-f002:**
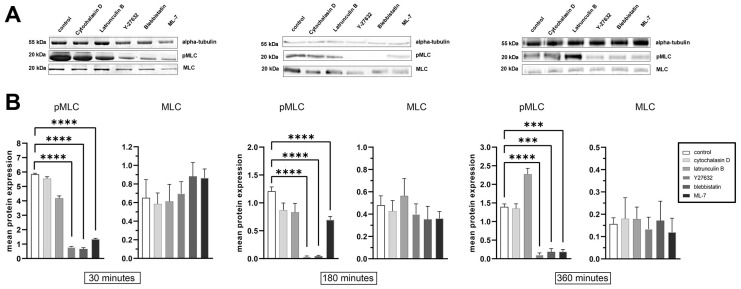
Representative Western blots (**A**) and quantification graphs (**B**) of phosphorylated myosin light chain (pMLC) and total myosin light chain (MLC) in U2OS cells treated with Y-27632 or blebbistatin for the indicated time points (30, 180, and 360 min). Densitometric analysis shows a significant reduction in pMLC levels, with phosphorylation at 180 min decreasing to 3% of the control level for both inhibitors. Alpha-tubulin was used as a loading control. Protein levels are represented as mean ratio values quantified from protein bands of pMLC versus total MLC, normalized to alpha-tubulin, and compared to untreated control cells. Data are shown as mean ± standard deviation (SD). Statistically significant differences were determined with the Kruskal–Wallis test with Dunn’s post test, *p* < 0.0001. (***)—*p* < 0.001; (****) *p* < 0.0001.

**Figure 3 ijms-26-07701-f003:**
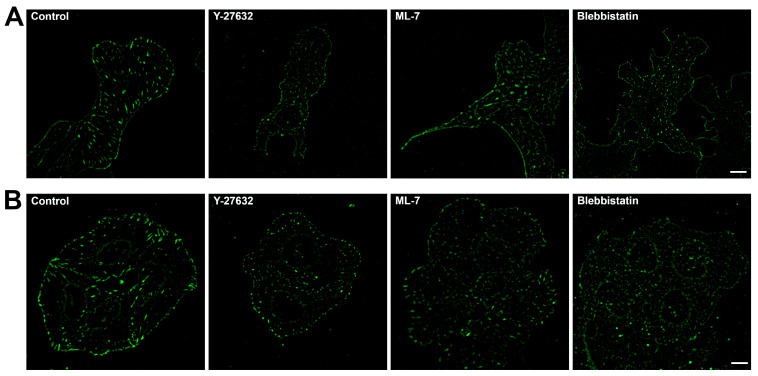
FAs in 3T3-vinculin-RFP and U2OS-vinculin-RFP cells after 30 min treatment with myosin II phosphorylation pathway inhibitors. (**A**) 3T3 cells, control, 30 min treatment with 10 µM Y-27632, 10 µM ML-7, and 45 μM blebbistatin. (**B**) U2OS cells, control, 30 min treatment with 10 µM Y-27632, 10 µM ML-7, and 45 μM blebbistatin. All figures are at the same magnification. Scale bar—10 µm.

**Figure 4 ijms-26-07701-f004:**
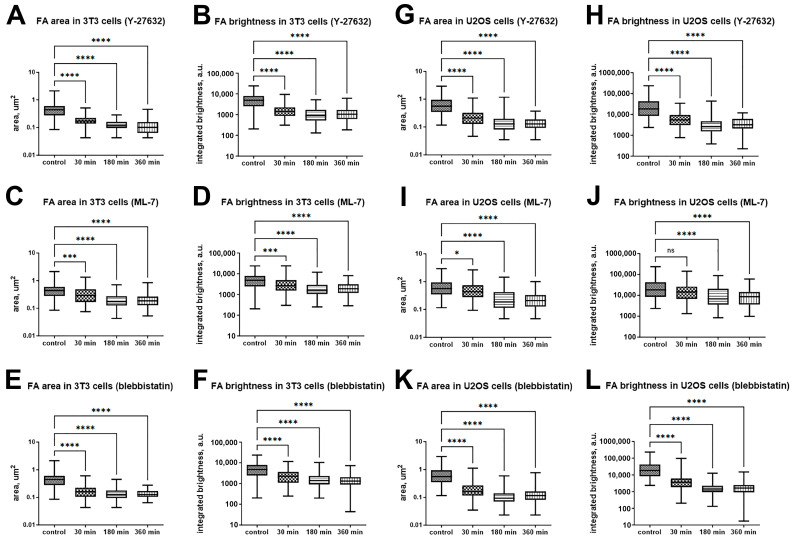
Quantitative analysis of the changes in FA area and brightness under the myosin II phosphorylation pathway inhibitors. FA area changes after treatment with 10 μM Y-27632 (**A**,**G**), 10 μM ML-7 (**C**,**I**), and 45 μM blebbistatin (**E**,**K**) in 3T3 and U2OS cells, respectively. FA-integrated brightness after treatment with 10 μM Y-27632 (**B**,**H**), 10 μM ML-7 (**D**,**J**) and 45 μM blebbistatin (**F**,**L**) in 3T3 and U2OS cells, respectively. Data are shown as median with range. (*) *p* < 0.05; (***)—*p* < 0.001; (****) *p* < 0.0001.

**Figure 5 ijms-26-07701-f005:**
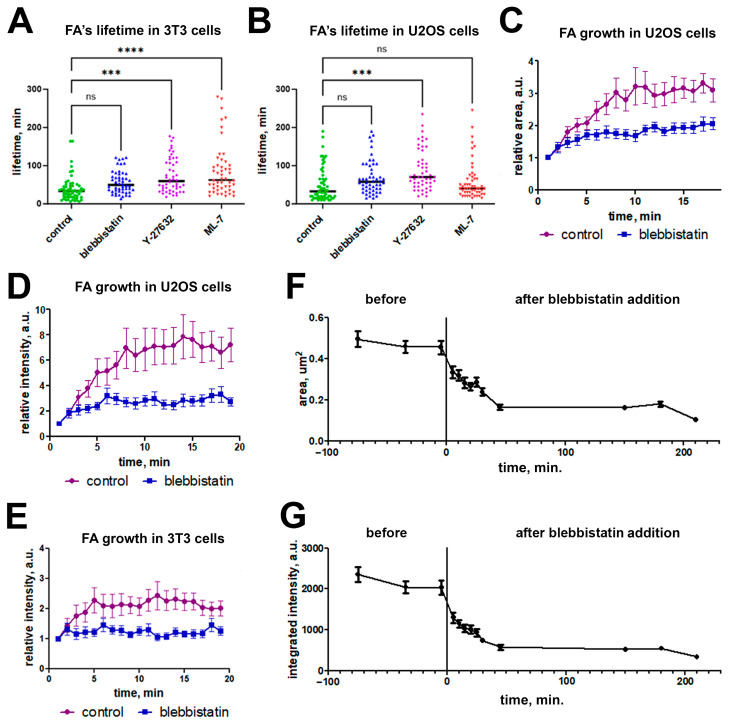
FAs under myosin II inhibitors. Dynamic parameters of FAs in control and under treatment with myosin II inhibitors. Median lifetime of FAs in 3T3 (**A**) and U2OS (**B**) cells. Downregulation of FA growth: area (**C**) and integral intensity (**D**) dynamics of FAs in U2OS cells under treatment with blebbistatin. (**E**) integral intensity dynamics of FAs in 3T3 cells. Area (**F**) and integrated intensity (**G**) dynamics during blebbistatin treatment of FAs in U2OS cells. Data on C-G are presented as mean ± SEM (n = 20). Vertical bar indicates time of blebbistatin addition. (***)—*p* < 0.001; (****) *p* < 0.0001.

**Figure 6 ijms-26-07701-f006:**
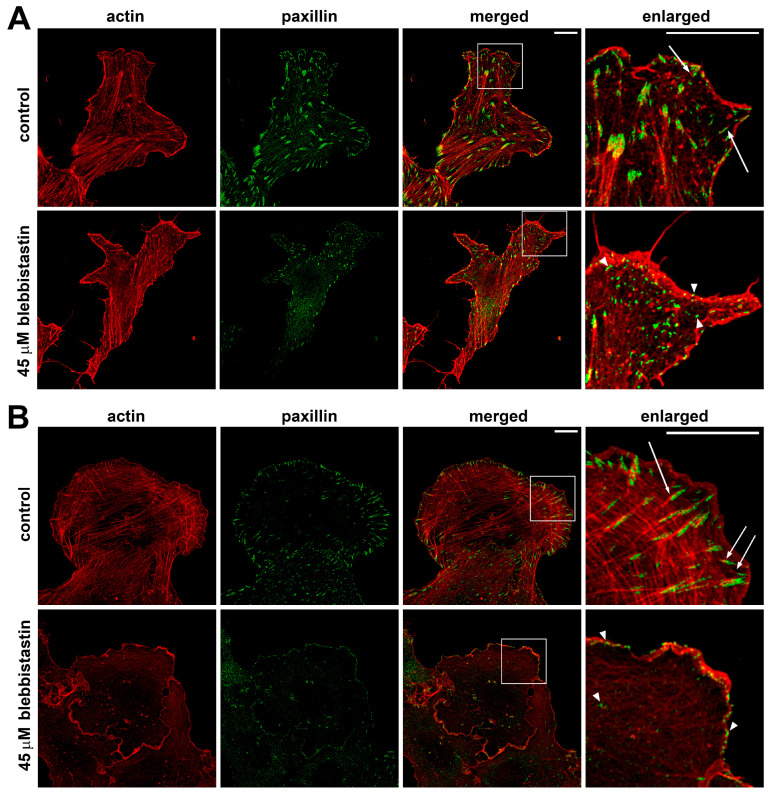
Actin and paxillin distribution in fixed 3T3 (**A**) and U2OS (**B**) cells. Paxillin (green) and actin (red) under normal conditions and when myosin II ATPase activity was suppressed by 45 μM blebbistatin. In the insets (4× magnification of the highlighted area), FAs are normally formed next to actin bundles (arrows); while in the presence of blebbistatin, actin bundles are not well organized near small FAs (arrowheads). Scale bars: 10 µm.

**Figure 7 ijms-26-07701-f007:**
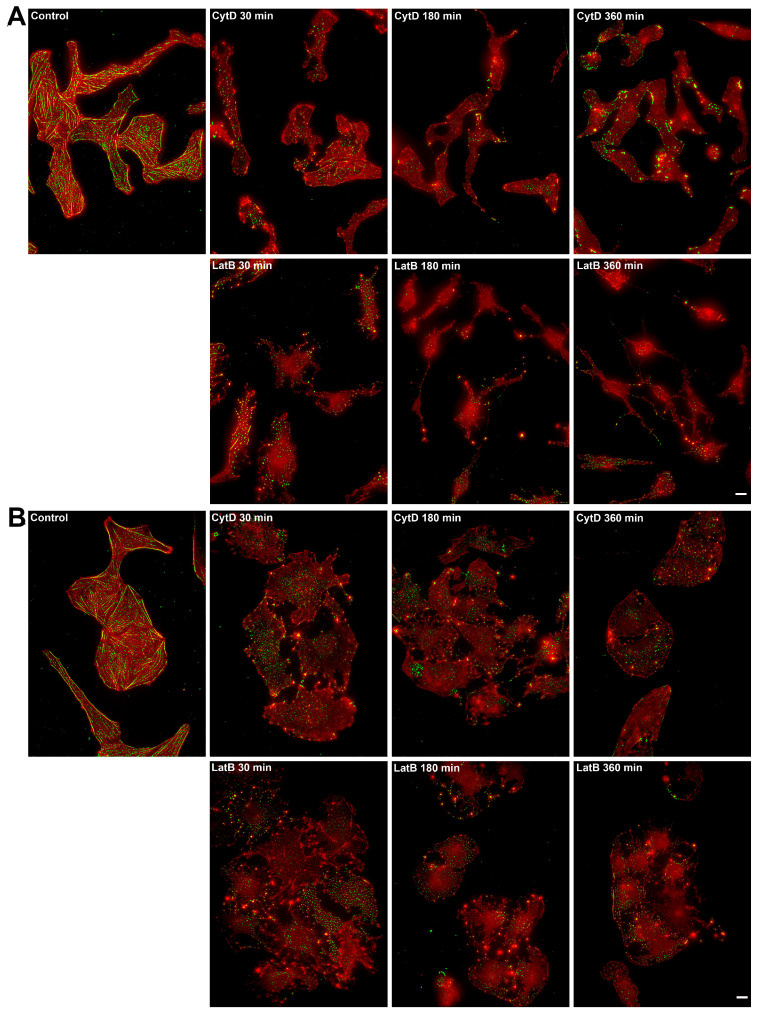
Actin and myosin II distribution under the action of actin inhibitors in 3T3 and U2OS cells. Alexa-Fluor647-phalloidin is in the red channel, pMLC is in the green channel. All figures are at the same magnification. Bar—10 μm. (**A**) In control 3T3 cells, the actin cytoskeleton is represented by numerous stress fibers, which are located perpendicular to the active edge and parallel to the long axis of the cell. Phosphorylated myosin II is associated with stress fibers. In the presence of 1 μM cytochalasin D, the actin fibers disassemble within 30 min. Phosphorylated myosin forms rather bright clusters in the cytoplasm that remained after prolonged treatment. In the presence of 1 µM latrunculin B, the F-actin structures are disassembled in 30 min, and phosphorylated myosin is reassembled into large clusters, which mainly disappear after prolonged exposure to the drug. (**B**) In U2OS cells, actin microfilaments form large ventral fibers in the cell body, thin dorsal fibers directed perpendicular to the active edge of the cell, and transverse arcs located parallel to the active edge of the cell. Phosphorylated myosin II in the form of small clusters is associated with all types of actin fibers. In the presence of 1 μM cytochalasin D, the actin cytoskeleton disassembles within 30 min; no changes are observed later on for several hours. Phosphorylated myosin II forms clusters at the cell periphery after 30 min of incubation with 1 μM cytochalasin D; over time, the number of phosphorylated myosin clusters decreases. In the presence of 1 μM latrunculin B, the disassembly of actomyosin complexes occurs in a similar manner.

**Figure 8 ijms-26-07701-f008:**
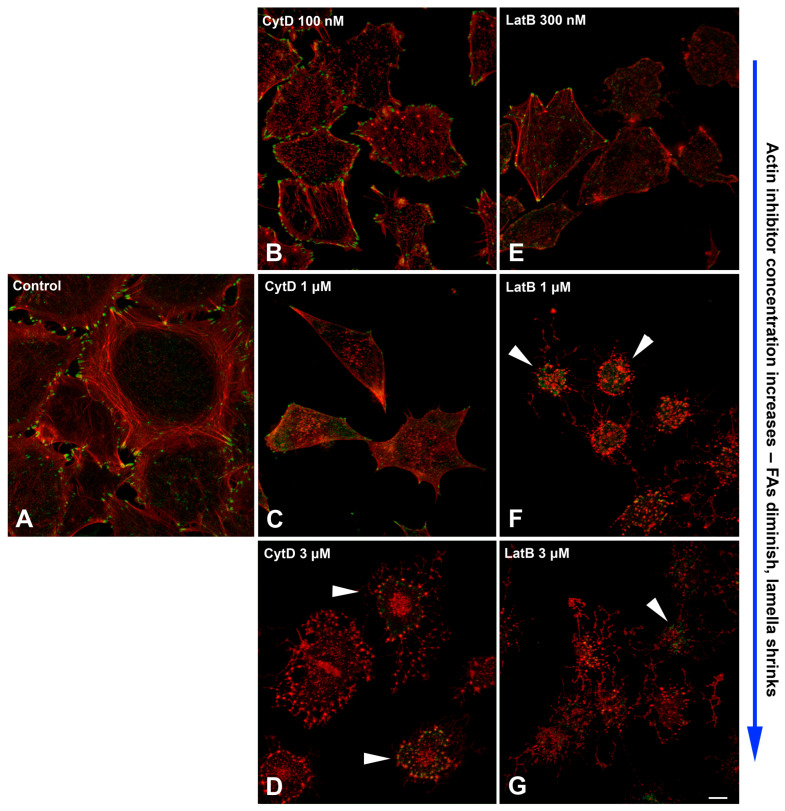
Actin and paxillin distribution in A549 cells under the treatment with actin inhibitors. (**A**)—control; (**B**)—cytochalasin D), 100 nM; (**C**)—cytochalasin D, 1 μM; (**D**)—cytochalasin D, 3 μM; (**E**)—latrunculin B, 300 nM; (**F**)—latrunculin B, 1μM; (**G**)—latrunculin 3 μM. Paxillin (green) and actin (red) in A549 cells in normal conditions and when actin polymerization is inhibited by increasing concentrations of cytochalasin D (100 nM → 3 μM) or latrunculin B (300 nM → 3 μM). LSCM, maximal intensity projections of ventral cell parts are shown. Arrowheads indicate cells with residual paxillin staining in the cytoplasm. All figures are at the same magnification. Scale bar—10 µm.

**Figure 9 ijms-26-07701-f009:**
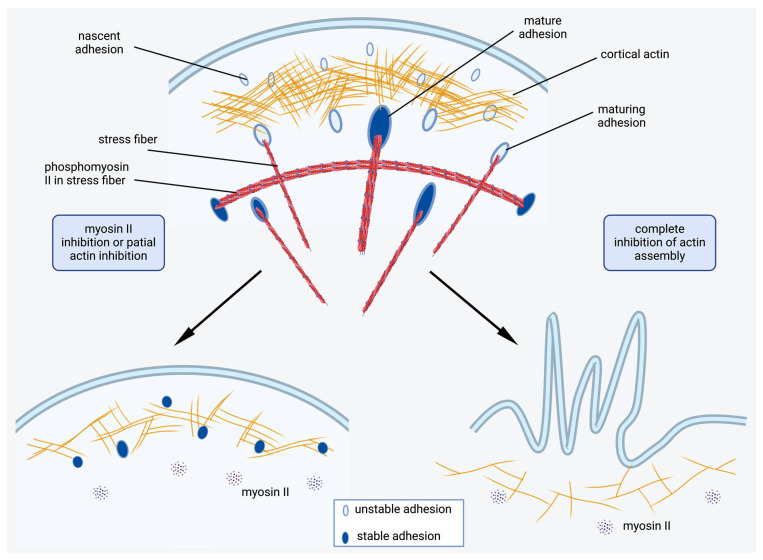
A model of FA regulation by the interplay between actin dynamics, actin polymerization, and myosin II-dependent tension. Under normal conditions of actin assembly and in the presence of active myosin II, the amount of major tension-dependent FA proteins (talin and vinculin) in nascent adhesions increases, and they can enlarge, becoming FAs. Under inhibition of myosin II phosphorylation, only cortical actin interacts with FA proteins; formation of small nascent adhesions is still possible. These adhesions cannot enlarge, but are rather stable in time. When actin assembly is impaired, FA proteins cannot interact with actin at the cell edge, thus no FAs form and destabilized lamellae shrink.

## Data Availability

All raw data supporting the results obtained are available from the authors upon request.

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
