# Peer review of "How Actin Polymerization and Myosin II Activity Regulate Focal Adhesion Dynamics in Motile Cells"

_ijms, 2025, doi:10.3390/ijms26167701_

Round 1
Reviewer 1 Report
Comments and Suggestions for Authors
This is a study designed examine the effects of Myosin-II and actin perturbations on the dynamics of focal adhesions. By using drugs to impact myosin II and actin function, the group looks at how focal adhesion are impacted. They discover that partial inhibition and complete inhibition of myosin II have different effects. Partial inhibition through either ROCK or myosin light chain kinase inhibitors leads to gradual focal adhesion shrinkage, whereas complete inhibition of myosin II resulted in disassembly of existing focal adhesions, followed by the formation of new, small focal adhesions at the cell periphery. Impacting actin through drugs like latrunculin B or cytochalasin D also promoted the formation of small focal adhesions.
The experiments are well-done, the conclusions are supported by high quality data, and the figures are easy to understand and appropriate for the content. This work will be of broad interest to people in the field of focal adhesion biology.
I only have minor questions/suggestions:
- Have the authors tried to impact actin using actin nucleator inhibitors? CK-666
- Have the authors tried looking at the effect of myosin II/actin perturbations on focal adhesions in these cells when the cells have been placed in a 3D matrix?
- Have the authors tried to apply stretching forces of cells that have been treated with myosin II/actin perturbation drugs?
Author Response
Please, see the attachment

Reviewer 2 Report
Comments and Suggestions for Authors
This study looked at how actin polymerization and myosin II activity work together to control focal adhesion (FA) dynamics in moving cells. Using live-cell imaging and drug treatments, the researchers found that partially blocking myosin II or actin polymerization shrinks FAs but actually makes them more stable, while fully blocking them leads to FA disassembly and lamella retraction. Interestingly, even when myosin II was completely inhibited, small new FAs still formed at the cell edges, and these small FAs lived longer than the normal ones. The experiments showed that actin polymerization at the cell cortex alone is enough to build small, stable adhesions, even when contractile forces are missing. However, if actin polymerization was completely stopped, FAs broke down quickly and the cells lost their spread shape. Overall, this work suggests that cortical actin assembly, not intracellular tension, is the minimum requirement for maintaining small focal adhesions and supporting cell attachment.
Major Concerns:
- The paper talks a lot about how intracellular tension affects focal adhesion (FA) dynamics, but there’s no direct measurement of the forces involved. Without tools like traction force microscopy or FRET-based tension sensors, it’s hard to firmly tie the FA behaviors to actual changes in tension. Adding direct force measurements would really make the conclusions more convincing.
- The study mainly uses chemical inhibitors like blebbistatin and Y-27632, but these can have off-target effects. It would be much more solid if the authors could use some genetic approaches too, like knocking down myosin II with siRNA or CRISPR. That way, they can be more confident that the effects they see are really due to myosin II inhibition, not something else.
- It would really help if the authors did a rescue experiment — for example, after inhibiting myosin II, they could reintroduce a phosphomimetic mutant of the myosin light chain to see if FA dynamics return to normal. That would make the causal link a lot stronger.
Minor Concerns:
- In Figure 2B, the phosphorylation levels should ideally be shown as the ratio of pMLC to total MLC, not just raw signal intensities. That would give a better sense of how myosin II activation changes overall.
- In Figure 8, it’s a bit hard to quickly tell which drug concentration caused full FA disappearance, because the images look pretty similar. I'd suggest clearly labeling each panel with the treatment (like "100 nM cytoD", "3 µM LatB") and maybe adding a quick summary chart or a cartoon to show the trend — for example, fewer FAs → lamella retraction.
- Since everything was done on 2D culture surfaces, it would be great if future work could test whether the same FA behaviors hold up in 3D environments, where tension generation and FA formation work quite differently.
- It would make the discussion stronger if the authors touched briefly on downstream signaling pathways, like FAK-Src, that might be affected by the changes they see in FAs.
Author Response
Please, see the attachment

Round 2
Reviewer 2 Report
Comments and Suggestions for Authors
The authors have addressed all major and minor concerns with appropriate detail, scientific rigor, and transparency. The additions and clarifications significantly enhance the manuscript. I support its acceptance pending any final editorial revisions.